



# TorchClim v1.0: A deep-learning framework for climate model physics

David Fuchs[1,4,*], Steven C. Sherwood[1,2,*], Abhnil Prasad[1,2,3,*], Kirill Trapeznikov[5,*], and Jim Gimlett[5,*]

[1]Climate Change Research Centre, Biological, Earth and Environmental Sciences, University of New South Wales, Sydney, Australia
[2]ARC Centre of Excellence for Climate Extremes, University of New South Wales, Sydney, New South Wales, Australia
[3]School of Photovoltaic and Renewable Energy Engineering, University of New South Wales, Sydney, NSW, Australia
[4]Climate and Atmospheric Science Branch, Department of Planning and Environment, Sydney, New South Wales, Australia
[5]STR, Woburn, MA, USA
[*]These authors contributed equally to this work.

**Correspondence:** David Fuchs (David.Fuchs@environment.nsw.gov.au)

**Abstract.** Climate models are hindered by the need to conceptualize and then parameterize complex physical processes that are not explicitly numerically resolved and for which no rigorous theory exists. Machine learning and artificial intelligence methods (ML/AI) offer a promising paradigm that can augment or replace the traditional parametrized approach with models trained on empirical process data. We offer a flexible and efficient framework, TorchClim, for inserting ML/AI physics surrogates that respect the parallelization of the climate model. A reference implementation of this approach is presented for the Community Earth System Model (CESM), where the authors substitute moist physics and radiative parametrization of the Community Atmospheric Model (CAM) with an ML/AI model. We show that a deep neural network surrogate trained on data from CAM itself can produce a stable model that reproduces the climate and variability of the original model, albeit with some biases. This framework is offered to the research community as an open-source project. The new framework seamlessly integrates into CAM's workflow and code-base and runs with negligible added computational cost, allowing rapid testing of various ML physics surrogates. The efficiency and flexibility of this framework open up new possibilities for using physics surrogates trained on offline data to improve climate model performance and better understand model physical processes.

## 1 Introduction

The ubiquitous approach to forecast weather and climate is with global circulation and climate models (GCMs). GCMs offer a coarse numerical grid representation of the climate system, with typical horizontal resolution of one hundred kilometers and a few dozen vertical layers. In this model design, the effects of unresolved meteorological phenomena such as boundary-layer turbulence, moist convection, water vapor condensation, and ice nucleation must be summarized as a handful of moments or other parameters calculated in each spatial grid location. Heat transport by radiation must also be calculated and depends on the details of cloud distributions which themselves must also be calculated. The process of introducing these quantities into a GCM is loosely termed "parametrization" (hereafter "traditional parametrization" or TP) and generally involves an arduous



development cycle where an often simplistic conceptual model representing each unresolved process is codified into the coarse model representation.

Despite decades of investment, climate models show systematic departures from observations such as erroneous rainfall distributions and sea surface temperature patterns (Masson-Delmotte et al., 2021). Different models also disagree on key aspects of our climate system and its future in a warmer climate, such as cloud feedback and climate sensitivity (Zelinka et al., 2020) and regional climate changes. Often, this disagreement is traced back to the parametrization of physical processes. For example, Fuchs et al. (2023b) showed that much of the disagreement among climate models in the positioning of the midlatitude jet could be attributed to differences in the parametrization of shallow convection. Likewise, much of the spread of low cloud feedback in models was directly attributable to their cloud parameterizations (Geoffroy et al., 2017) or to sometimes spurious convective behavior (Nuijens et al., 2015). In some cases, model errors were linked to difficulties in tuning these parametrizations (e.g. Schneider et al., 2017, and references therein). Yet in some cases, the sheer number of parametrizations and versions of parametrizations in current GCMs could point to a fundamental problem with TP (e.g. for a list of parametrizations of moist convection, refer to Fuchs et al., 2023b).

The difficulty of developing TP, and increasing computational power, have led to growing enthusiasm for very high-resolution global models where it is hoped that processes such as convection and clouds can be explicitly represented by the equations of motion (Satoh et al., 2019). While this is an exciting development, such models are many orders of magnitude slower than traditional GCMs and therefore cannot replace standard models for most purposes. Moreover, even the highest resolutions contemplated will still require parameterizations of some processes (e.g., microphysics and turbulence). The current effort is motivated by the evident need to improve physics parameterizations, particularly for models run at affordable grid sizes.

Improving a TP involves substantial intellectual and engineering effort, requiring first the introduction of a new conceptual representation of partially observed processes that is parsimonious and yet captures all features thought to be essential, and then the substantial engineering challenge of codifying it into a GCM. In many cases, the development of new ideas has been hindered by computational complexities. For example, CAM version 5 introduced new features and increased the complexity in existing parametrizations that degraded the computational performance fourfold compared to CAM version 4. In general, GCMs suffer from the curse of dimensionality (Bellman, 1966), and for many years the increase in the computational complexity of GCMs was met by Moore's Law in its various forms (Wikipedia, 2022). The increase in computational complexity of GCMs was met by a matching increase in hardware performance (CPU, network latency and throughput, storage speed and capacity, etc.). This helped keep model performance at acceptable levels in the face of the increase in model complexity, including more processes and components, more elaborate parametrizations, and finer numerical grids. Unfortunately, recent years saw the saturation of Moore's Law in its various forms.

Despite this, recent years have seen an increase in distributed storage and computing capacity and, notably, the repurposing of the graphical processing unit (GPU) for data processing and ML/AI. These developments have helped bring a renewed interest in inverse modeling and machine-learning approaches. One approach that has proven useful for weather forecasting is to replace the entire atmosphere model with an empirical one (Bi et al., 2023). Here we do not consider this option, but rather,





the use of a *hybrid model* which uses empirical learning to improve parameterizations or replace them with a *physics surrogate*. One way of doing this is to use observations to tune existing or new parametrizations (e.g. Dunbar et al., 2021; Howland et al., 2022; Schneider et al., 2017). This approach replaces the manual tuning step in parametrizations, addressing uncertainties associated with manual tuning. However, these approaches rely on a small set of parameters, which themselves are manually
chosen, and a fixed set of possibly flawed structural assumptions in the parametrizations. Kelly et al. (2017) replaced physical parameterizations with a linear tangent model fitted to results from a process model but obtained disappointing results even in the tropics where variations are relatively small, presumably because the actual physics are too nonlinear.

A few approaches have been tried for developing nonlinear empirical physics surrogates. One is to train a new parametrization from a more complete set of data but leave it bound to the input data. O'Gorman and Dwyer (2018) used a random forest
algorithm as a drop-in replacement of moist convection to emulate the original scheme, while Yuval and O'Gorman (2020) used a similar approach to represent the impact of unresolved motions in a much higher-resolution training simulation. This data-bound approach has the advantage of being able to obey conservation laws and properties of different variables (e.g. by design, precipitation rate cannot be negative). However, a random forest algorithm is unlikely to extrapolate successfully beyond the input data, as the authors found when trying to simulate warmer climates.

An approach that might hold more promise for extrapolation is the use of deep neural networks (NNs or DNNs). NNs have been used extensively in climate science and have seen a growing interest as an alternative to TPs. For example, Brenowitz and Bretherton (2019) trained a DNN using data from a 4 km spatial resolution near-global aqua-planet cloud-resolving model (CRM). This was used to learn heating and moistening tendencies in a coarse-grained 160 km resolution representation of the same model, serving as a drop-in replacement to the original parametrization. This model was able to run for a few days before
becoming unstable. Wang et al. (2022) train a DNN using a super-parameterized (SP) GCM and use the trained model in a non-SP version of the same GCM. They report a significant boost in computational performance compared to an SP GCM, alongside the ability to reproduce features from the SP parametrization.

Despite considerable interest and availability of data and computational resources, NNs are yet to find their place in climate physics parametrization. NNs are mostly used in offline test beds, simplified GCMs, and scenarios with limited boundary con-
ditions such as aqua-planet simulations. Model stability has been a key issue preventing progress, with a myriad of approaches proposed to diagnose and correct hybrid GCM-NN model instabilities (e.g. Brenowitz et al., 2020, and references therein). In many cases, stability issues have been linked to the NN model learning non-causal relationships or the hybrid GCM-NN model being unable to obey conservation laws. For example, Brenowitz and Bretherton (2019) attributed instabilities in their model to non-causal process influences from the upper troposphere that their DNN learned. They fixed the issue by manually removing
inputs to their DNN above 10km. Wang et al. (2022) performed an extensive manual model search to learn a DNN that results in a stable hybrid model. They also identified missing variables in previous studies, such as direct and diffuse radiation at the surface, which are required by surface models. However, their model did not predict precipitation components, such as convective precipitation and snow. These are required by the land surface model and are likely to be required in order to run a stable fully coupled climate scenario.



Aside from stability, the predictive skill of NNs and DNNs has also been an issue. For example, Mooers et al. (2021) found that their DNN showed selected regions with low skill on the 15-minute time-step of their data while exhibiting high skill on coarser temporal resolutions. This issue was traced to the difficulty of the DNN to emulate regions with fast stochastic signals, such as tropical marine boundary layer convection. Others found that their NNs struggled in the stratosphere (e.g. Gentine et al., 2018; Brenowitz and Bretherton, 2019). This issue seems to be related to the lack of moisture in these levels. Aside from

skill and stability issues, this aspect could affect the ability of these models to emulate a climate with an increase in tropopause height.

Other approaches alleviated the lack of skill and stability issues either at run-time or during an offline learning stage. Rasp (2020) used an online learning approach, where the DNN was supported by a numerical model that "advises" the ML model. In this case, it was less important to achieve a perfect DNN, to the extent that it perfectly obeys conservation laws and causality.

However, it raises questions as to the extent to which the DNN can deviate from its online advisor. Beucler et al. (2021) added linearized constraints to the learning loss function to help focus the learning. Others augmented the loss function with Boolean loss terms, taking into account the binary nature of precipitation, penalizing errors in the presence or absence of nonzero precipitation (Cannon, 2008; Rampal et al., 2022). Mooers et al. (2021) found complementary improvements in skill from constraints and hyper-parameter tuning. Yet another challenge involves the ability to characterize the learned function that the

DNN produced, for example, to interpret errors and instabilities. While the literature offers extensive discussion on the topic, for many approaches it is hard to attach a meaningful physical interpretation. Brenowitz et al. (2020) propose an approach that diagnoses the stability of an atmospheric model by using lower tropospheric stability and tropospheric moisture as stability indices. Their approach has the advantage that it is easy to implement and has a clear physical interpretation in low dimensions.

Yet despite these advances, we are yet to see a "production-ready" DNN-based parametrization. Beyond stability issues, the

use of empirical physics surrogates faces several challenges. First, the approach needs a long and fully resolved observational data sample. While using high-resolution CRMs solves part of the issue, CRMs are also partly parametrized and tend to produce short or regional datasets. Consequently, it is desired to have an approach to blend inputs from multiple sources. Second, a key advantage of ML-based parametrizations is the ability to bypass stages in the development of TP, thus shortening the development cycle. However, this aspect is yet to be demonstrated. It is also required to embed tests or guarantees of stability

within the development workflow. Third, a true implementation of this approach will likely require accommodating multiple ML models and model architectures, some of which might need to coexist within a single run.

Another promise of such a framework is faster and cheaper computation compared to TPs. However, current implementations are scarce and tend to either be hard-coded into the GCM or based on a bridge pattern (Zhong et al., 2023; Wang et al., 2022, e.g.), where a GCM is linked to remote GPUs in the sense that it crosses process or network boundaries (Fig. 1). Such an

approach likely cancels the scalability benefits of GPU, due to the data-transfer overhead over media that is many times slower than CPU access to local memory. Moreover, since most current GCMs are based on CPUs, the GPUs will be idle during much of the GCM execution (and vice versa for CPUs). An approach that allows a pure CPU-based implementation and does not break the parallelization model of the GCM, while allowing for a smooth transition to GPUs if desired, would be far preferable.





Ideally, such a framework should enable surrogate parametrizations to be developed through the use of interpreted languages,
which can accelerate development.

Hence our main objective is to offer a framework that would facilitate the flexible introduction of ML/AI surrogate models
into a GCM. Section 2 presents the features of this framework and its current scope. This is followed by Section 3 where
a reference implementation is added into CESM/CAM. Section 4 presents an evaluation of this framework, while Section 5
discusses desired extensions to the framework.

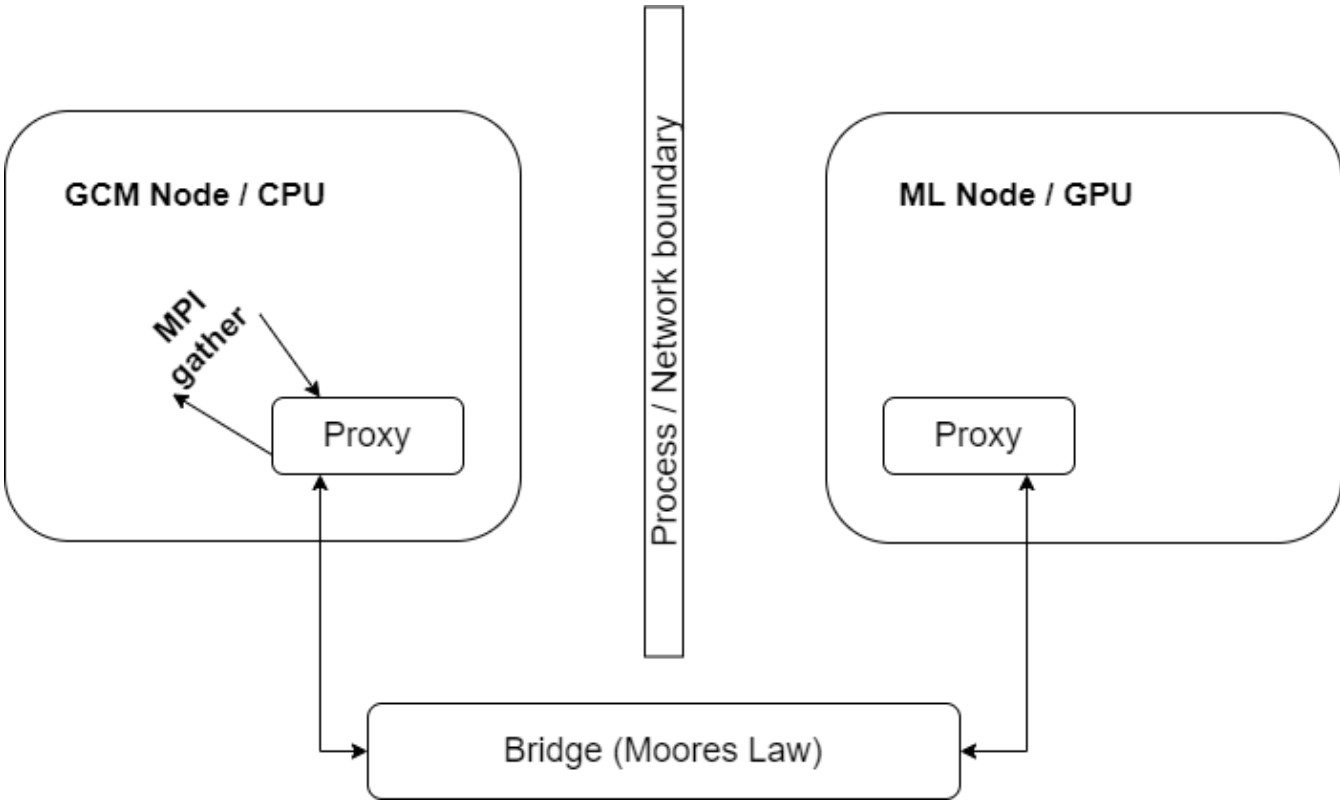

**Figure 1.** An illustration of the key aspects of an implementation of a hybrid NN-GCM through the use of a bridge pattern. The GCM
interacts with the NN parametrization, which resides on a remote GPU. This requires crossing process (memory) and network boundaries
which are many times slower than random access memory, thus canceling the scaling benefits in using a GPU.

## 2   TorchClim Framework


The discussion so far suggests that, while promising, ML/AI surrogates have yet to find their place as drop-in replacements
of parametrizations in existing GCMs, due to scientific and engineering gaps. It is therefore desired to offer a framework by
which ML/AI-based models are added to GCMs. We also have noted a diverse range of use cases of ML/AI, highlighting the
possibility that these approaches can help shorten the development cycle of new parametrizations. We seek to facilitate this





exploratory quality via the proposed framework. In particular, a situation is within grasp where many ML/AI models might be studied, while protecting the investment in existing CPU-based GCMs, and without compromising the potential of future shift of GCMs to GPUs or other technologies later on.

## 2.1 Requirements

To achieve this, the following set of requirements and features are implemented via the proposed framework:

1.  It can be readily adapted to replace any portion of the GCM, focusing on (but not limited to) physics parameterizations;

2.  Offers a concise and scalable design pattern, combining ease of use and run-time performance;

3.  Offers a rapid "plug-and-play" replacement of previous physics surrogates with new ones;

4.  Allows multiple ML/AI models to coexist, in the same run or as an overlay on top of TPs, for example to replace two different TPs or for data blending, online learning, or side-by-side with TPs;

5.  Allows ML/AI parametrizations to coexist in the same source code branch, and execution flow of a GCM alongside TP approaches;

6.  Uses existing parallelization frameworks and infrastructure (i.e. MPI over CPUs), but without limiting the ability of ML/AI approaches to use GPUs on demand inside a GCM or during the learning stage;

7.  Allows ML/AI approaches to be activated under specific conditions, for example in a given time period and region;

8.  Offers a workflow and supporting tools to boost the learning process of ML/AI models—therefore, since most GCMs are written in Fortran, a Fortran interface in the reference implementation;

9.  Allows the use of scripted languages during the learning process while avoiding the need to implement an ML/AI model in Fortran.

TorchClim bundles these requirements into a lean framework that is licensed under GPL 3.0 and include a detailed user and 155 installation instructions (Fuchs et al., 2023a). Our approach relies on PyTorch (Paszke et al., 2019). PyTorch is a deep-learning neural network framework that offers both Python and C/C++ interfaces. As such, it meets the requirement to expose an interface to both scripted and compiled languages. Therefore the learning cycle can be enhanced through the use of a scripted language while climate models can natively access byproducts.

## 2.2 Architecture overview

The implementation of TorchClim is broadly divided into two main components: the first encapsulates the details of the ML/AI framework, hiding the details of the interactions with the underlying implementation (LibTorch) from the GCM, while the second encapsulates the interactions with the GCM (Fig. 2). The first component is designed as a plugin that is loaded at



runtime by the GCM. Given that most GCMs are written in Fortran, this component defines (but is not limited to) a Fortran interface, and can support any number of ML/AI models (Fig. 2, lower block of modules). Generally, this component accepts

a set of prognostic variables that the GCM supports.

At runtime, an instance of this plugin lives within the scope of a thread local storage (TLS) inside an MPI rank. The plugin handles initialization and configuration of LibTorch, loading and calling ML/AI models, and if required, keeping state and alignment between MPI ranks and GPUs, and stateful ML/AI models (see Section 5). It also takes care of packing variables according to the input and output specifications of the underlying ML/AI model. All that is bundled in a shared library that

is agnostic of the GCM that is calling it. An advantage of this approach is that it allows for an intermediate validation step between the scripted training phase and the final stage, where the ML/AI model is used in a GCM.

The second component encapsulates the details of a specific GCM. It is in charge of coupling the ML/AI inputs and outputs into a specific GCM workflow (Fig. 2, upper block of modules). This component is exposed to the GCM as one or more standard parametrization modules. This component can serve as a drop-in replacement of the portion of the parametrization that will be

replaced by surrogate ML/AI models. The general workflow of this component is to pull state information from the GCM, and interface the first component, invoking a surrogate ML/AI model. It then retrieves predicted outputs from the surrogate model and plugs them back into the GCM workflow. With the intention that this framework be used as a replacement of physics parameterizations, it is expected that these modules will ingest prognostic variables such as temperature, specific humidity, condensed species, aerosols, etc, and output learned tendencies that would be incorporated back to the GCM's workflow.

## 2.3   Parallelization framework

We note that most GCMs rely on CPUs and scale via MPI/MP discretization. However, ML/AI frameworks tend to scale better when using GPUs. A key prerequisite in our implementation is to prevent duplication of parallelization frameworks inside the GCM and work along the geographical chunking of the GCM's physics. Other approaches mostly relied on the bridge pattern, which generally means a blocking step inside the climate model that gathers the global state and sends it to a remote

GPU over dedicated infrastructure. As discussed earlier, this breaks the parallelization strategy of the GCM in the sense that it brings a second parallelization framework (the GPU) to run alongside the GCM. Our approach avoids the bridge pattern and parallelizes the ML/AI model along the MPI discretization. The user can still choose to run the ML/AI model on the GPU, but this runs locally (within the MPI rank). As depicted in Fig. 3, many GCMs divide the geographical domain into tiles, with each tile hosted by an MPI rank. TorchClim's GCM-independent plugin creates an instance of LibTorch in the thread local

storage (TLS) of each rank and loads the necessary ML/AI models into it. The benefit of this approach is that the ML/AI plugin behaves like any other module or package in the GCM, without requiring architectural changes. This helps protect the vast investment in CPU-based GCM implementations. The ML/AI model can be called over CPU or GPU, with the former having the option to further parallelize by hosting a thread pool per MPI rank (Fig. 3). These details are set via a configuration option inside the TorchClim framework.



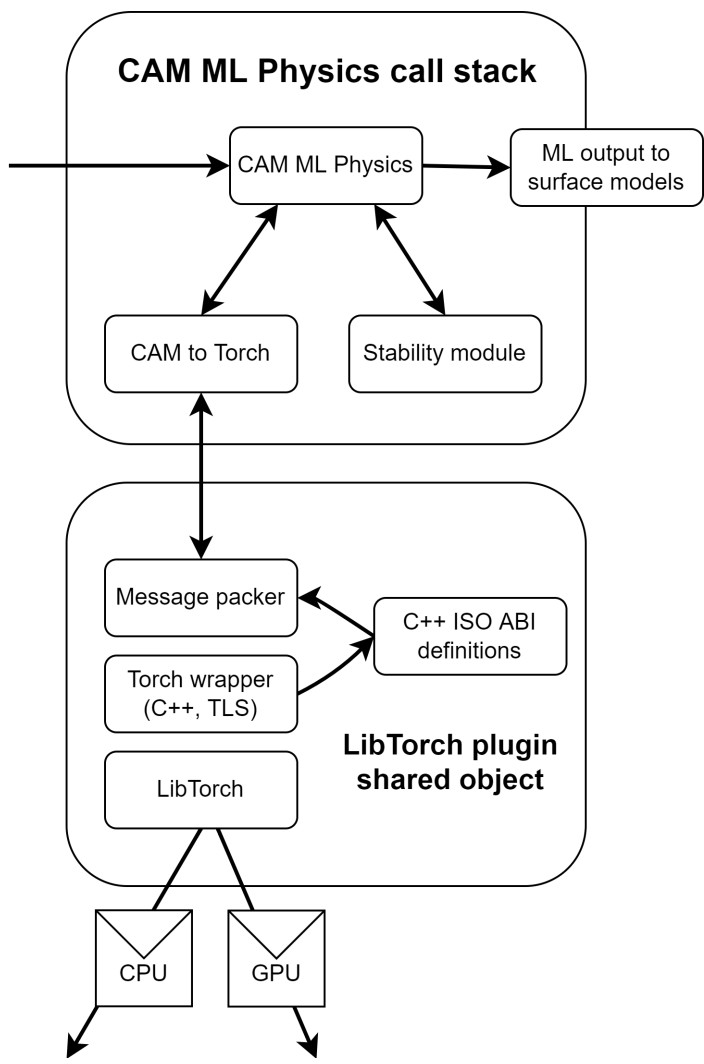

**Figure 2.** An overview of key components of the call stack. The top shows the component that connects a GCM's physics workflow and the bottom shows the GCM-independent plugin that connects to ML/AI models. Notably, this plugin allows the user to direct calls to a local GPU or CPU.

## 3  CAM Implementation

We provide a reference implementation for the Community Atmospheric Model (Neale et al., 2010) that is part of CESM version 1.0.6, offering a drop-in replacement surrogate model for the sum of moist and radiative physics TPs. The surrogate model is a DNN that emulates the total tendencies of moisture and heat due to moist processes (convection, clouds, boundary layer) and radiation. Other parameterizations such as eddy diffusion, gravity wave drag and CAM's dynamical core are left running and are not emulated.




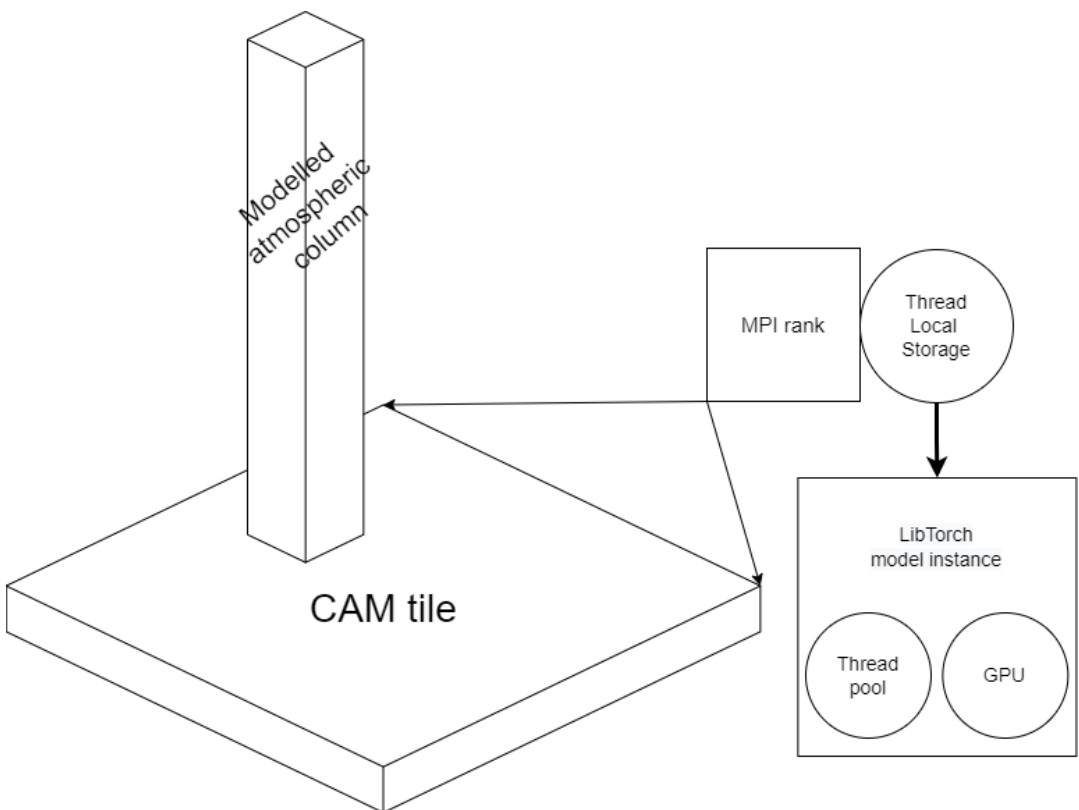

**Figure 3.** An architecture diagram depicting the relationship between a GCM's physics geographical tile (chunk), representing a set of adjacent geographical grid locations, and a LibTorch instance. In a GCM with MPI support, both coexist within the MPI rank. LibTorch ML/AI models can subsequently execute on a local GPU or a thread pool.

Like many other models, CAM relies on the Message Passing Interface (MPI) to discretize its domain, dividing the workload across processes and compute nodes. The physics parametrization suite in CAM divides the geographical domain into tiles of adjacent grid locations, with each tile associated with an MPI rank (process). Each grid location represents an atmospheric column, in which the physics runs independently from other columns. This dictates the required spatial dimensions of the input and output of the surrogate model. Our choice to replace the sum of moist and radiative physics parameterizations further dictate the set of input variables the surrogate model will ingest and the outputs it will produce. These are specified in Table 1. Generally, such substitution of physical parametrizations impose a set of minimum output requirements that the surrogate model needs to comply with. Here we divide them into three categories:

1. Tendencies that are required by the physical parametrization downstream from the point of insertion of the ML surrogate. For example, heating tendencies in our reference implementation.



2. Variables required by other downstream models. For example, surface models use convective snow rate in our reference implementation.

3. Variables required by the model dynamics. For example, cloud ice in our reference implementation.

Not complying with each of these types of interfaces could lead to instabilities at run-time or to an inferior surrogate model. For example, Wang et al. (2022) found that direct and diffuse short-wave radiative variables must be predicted by their surrogate model. This is not surprising given that in CAM downstream surface models require that these variables be provided. Likewise, we note that our setup left out cloud liquid and ice condensates.

**Table 1.** CAM history output variables used to learn the ML/AI model. The input variables contain CAM prognostic variables and atmospheric boundary conditions, while the output dimension contains variables that are needed to comply with CAM's interface downstream from the point of insertion of the NN model. Variables that are marked by '*' are used for diagnostics and are not essential at run-time.

|  | Inputs | Outputs |
|---|---|---|
| Profiles | $Q_v$(z), T(z), U(z), V(z), OMEGA(z), Z3(z) | $\dfrac{\partial T(z)}{\partial t}$, $\dfrac{\partial Q_v(z)}{\partial t}$ |
| Scalars | PS, TS, SOLIN, SHFLX, LHFLX, LANDFRAC, OCEANFRAC, ICE-FRAC | FSNS, FLNS, FSNT*, FLNT*, FSDS, FLDS, SRFRAD, SOLL, SOLS, SOLLD, SOLSD, PRECT*, PRECC, PRECL, PRECSC, PRECSL |

### 3.1 Alternate physics workflows

We recognize the potential of ML/AI to serve a range of use cases. For example, users may wish to run an ML/AI surrogate only at a certain time step or within a particular geographical region. Alternatively, it may be desired to run multiple models side-by-side or as a replacement for different physics parameterizations. For example, this could be to compare them for research purposes or to use TP to error-correct the ML parametrization. The reference implementation supports this need by placing a *mode selector* before the call to the physics parametrization (Fig. 4). An advantage of this approach is that it offers users a way to extend CAM with new capabilities without removing existing ones. The reference implementation still offers the original parametrization of CAM, alongside an ML/AI mode that runs combined moist and radiative parametrizations as an ML/AI surrogate, and a dual mode that runs both of these side by side. Our implementation allows the user to achieve that out-of-the-box (without directly changing CESM/CAM code-base).

### 3.2 Assisting the learning phase

So far, we assumed that an ML surrogate is available, without addressing where the data that would be used to learn it would come from. We note that a good starting point for training a surrogate model, as implemented in many previous studies (see Section 1), is simply the TP of the host GCM itself. In the reference implementation, this is the moist-physics and radiative





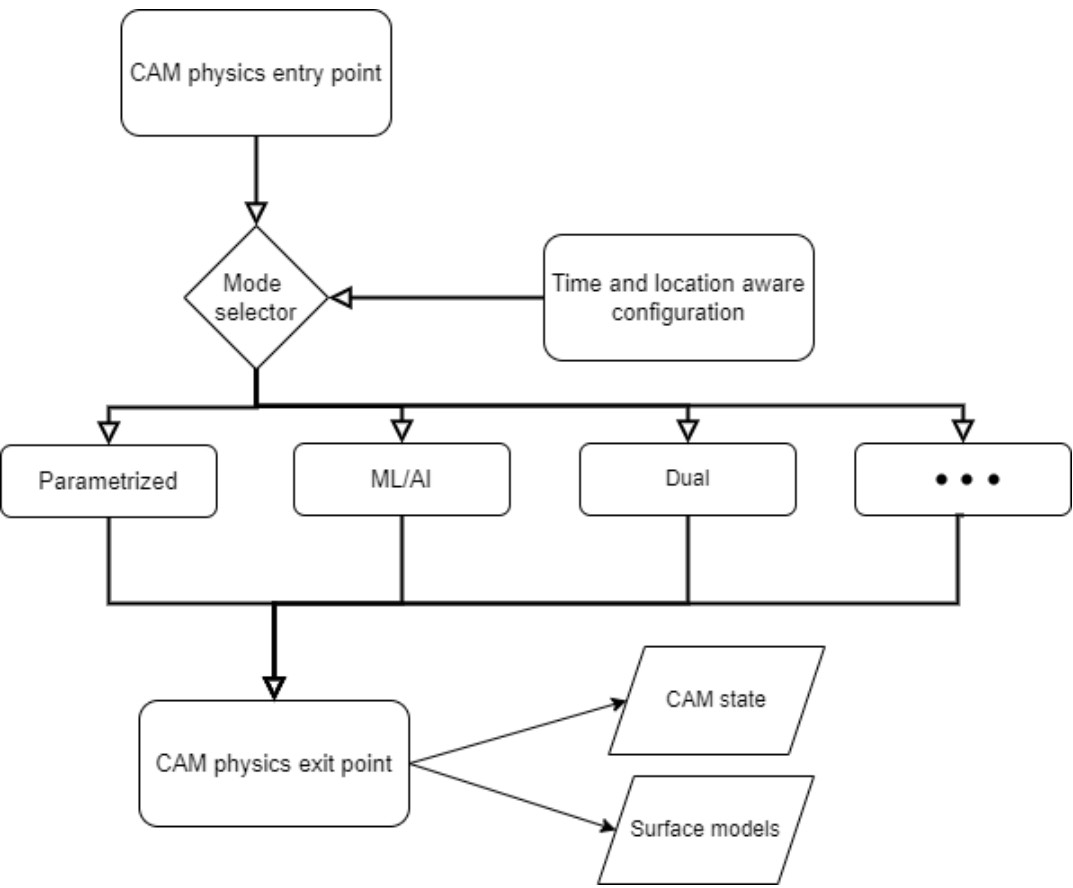

**Figure 4.** A sketch of the *mode selector*, which allows the reference implementation to add new models of CAM physics alongside the original parametrization. The *mode selector* is placed before the entry point to CAM's physics parametrization, allowing the selector to choose the workflow for a given spatial and temporal location and use case. The selector facilitates fast turnover to extend CAM with additional physics workflows without compromising other workflows.

parametrizations of CAM. To this end, the reference implementation of TorchClim is shipped with the *export_state* Fortran module. The user is provided with two subroutines: the first is inserted in the TP before the location where the surrogate model is to be called, while the second is inserted after that point. For example, our reference implementation replaces the 235 parametrizations of moist physics and radiation, so these subroutines are placed in the TP before and after these parameterizations (Fig. 5). These add additional history variables to CAM's output, recording state and accumulated tendencies before and after the desired section of TPs.





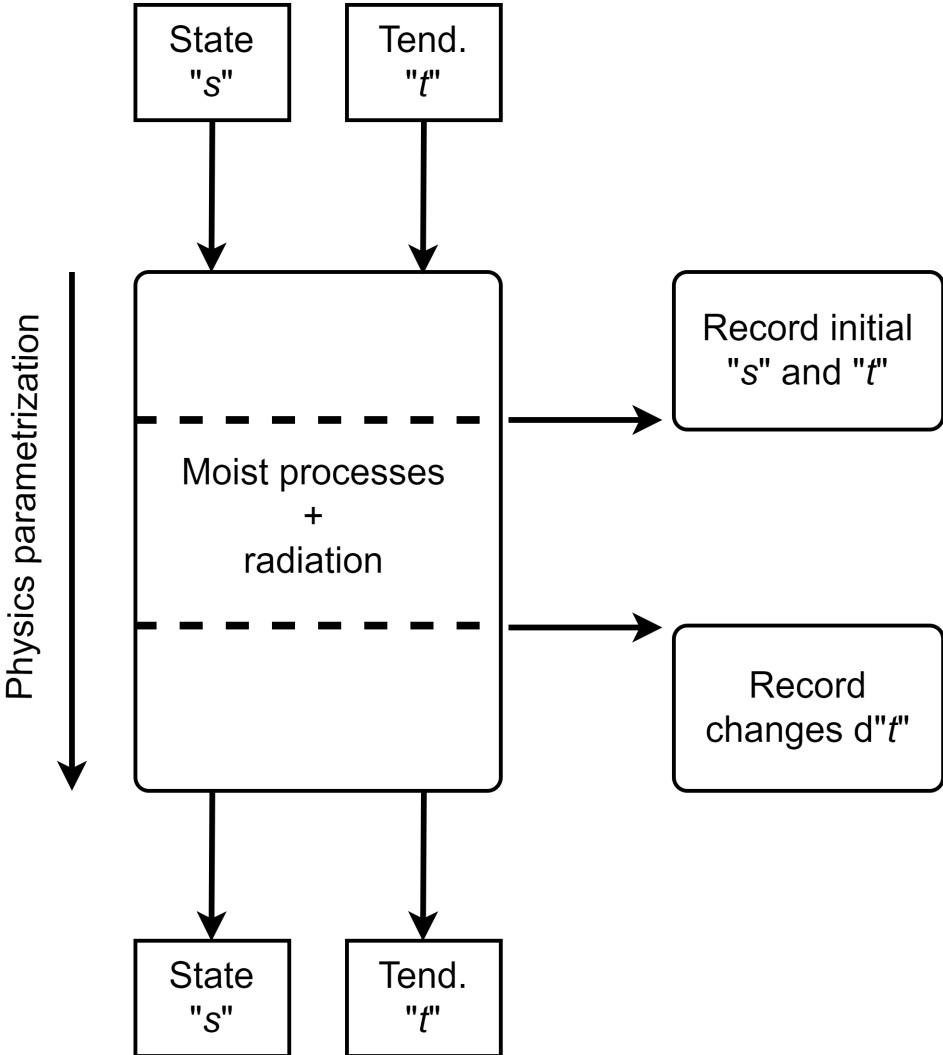

**Figure 5.** The process of recording the state and tendencies in a TP case that will be used in the ML/AI learning process. Here we exemplify it using the TP of moist and radiative parametrization in CAM. The *export_state* module allows the user to take snapshots of CAM's physics state and tendencies before and tendencies after these TPs. These are saved to CAM's history files ad subsequently used to train a baseline DNN (e.g. for our reference implementation).

## 4 Evaluation

We now evaluate the end-to-end process of outputting training data (described in Section 3.2), training a DNN offline using these data, and then deploying this DNN as a surrogate inside CAM using TorchClim. In this case, success is evaluated on the computational performance and accuracy of the hybrid model relative to the original CAM, whose physics the DNN is emulating.






Since our reference implementation allows for both TP and hybrid versions of CAM's physics to coexist under the same code base, this version of CAM is used to produce the training dataset. We generated a ten-year run with this version of CAM

using monthly AMIP SSTs (from 1979 to 1989). This run was configured to call the radiative parametrization at every model time step rather than hourly in the original configuration. Likewise, it produced history outputs at every physical time step, allowing the training data to be composed of adjacent time steps for the inputs and outputs of the DNN. History variables were written for the state before and after the insertion point in CAM's physics parametrization using the framework described in Section 3.2.

The first nine years of this dataset were used for training and validation, and the last was used for testing. Training samples were drawn uniformly over space and time while testing data is sampled over time only to preserve spatial information. The training sample size is proportional to the size of the time-space dimensions and matches $size = [365 * years * longitudes * latitudes]$ or $365*9*144*96 = 45e+6$ instances. This dataset was also randomly divided into 90% training and 10% validation datasets.

**4.1 DNN model description**

The DNN was trained offline via PyTorch's Python interface. The input dimension of the DNN contains CAM prognostic variables and atmospheric boundary conditions, while the output dimension contains variables that are needed in order to comply with CAM's interface downstream from the point of insertion of the DNN model. The CAM inputs and output variables that the DNN used are defined in Table 1. These are composed of scalars such as radiative fluxes and vertical model level profiles

(26 levels in this version of CAM).

The DNN is composed of seven fully connected layers. Each layer (except for the last) is followed by a batch normalization, dropout, and a linear rectifier. We use zero-mean / one-standard-deviation normalization per feature before the first layer, i.e., separately normalizing each model level (z) globally for each 3-D input variable, such as water vapor and each 2-D variable (Table 1). Correspondingly we use one-standard-deviation denormalization after the last layer for denormalizing the output

during inference.

The MSE on the normalized output variables is the initial loss function to be minimized. We also introduce additional terms in the loss function to address the biases described below. We add L2 regularization with a weight of one on all parameters except the biases and batch norm. Several constraints on range, equality and conservation to prevent unphysical predictions are encoded as either parameterization of the outputs in the DNN model or as additional regularization terms

in the optimization objective. (see Section 4.2 for more details). The optimization loss function consists of several terms: $L_{total} = L_{MSE} + \alpha_{L2}L_{L2} + \alpha_{eq}L_{eq} + \alpha_{cons}L_{cons}$. Loss trade-off coefficients $\alpha_{(.)}$ are chosen experimentally. Optimization is done using the Adam method (Kingma and Ba, 2014) with a learning rate of 1e-3. We train for 100 epochs, with a large batch size of 24 x 96 x 144 atmospheric columns that are randomly selected in time and space from the original ten-year dataset. Finally, we use linear warm-up for the first 10% of epochs and cosine cool-down learning rate scheduler to zero.

During training, a separate validation set is used to monitor optimization progress and perform early stopping if necessary.



In practice, we tested dozens of versions of the DNN with various bug fixes and tweaks. This was made easy by the Torch-Clim interface, which enables newly trained DNNs to be dropped in with no recompile of CAM required.

## 4.2 Constraining the target solution

Past studies and our efforts have found that to obtain good performance it is necessary to apply physical constraints to ML
surrogates to prohibit nonphysical predictions (Beucler et al., 2021; Brenowitz et al., 2020; Mooers et al., 2021; Karniadakis et al., 2021). In training the model for our reference implementation, we found a set of rules that helped constrain the target solution. Here we propose a classification of such constraints into three types according the nature of the applied constraint (Table 2).

**Table 2.** Classification of constraints of the solutions' space.

| Type | Name | CAM Relations | Implementation |
|------|------|---------------|----------------|
| Type I | Range constraints | $PRECT, PRECC, PRECL \geq 0$ | output param. |
| | | $FSDS, SOLL, SOLS, SOLLD, SOLSD \geq 0$ | $max(x, 0)$ |
| Type II | Redundancy constraints | $SOLL + SOLS + SOLLD + SOLSD = FSDS$ | residual reg. |
| | | $PRECC + PRECL = PRECT$ | $\|x_j - \sum_{i=1}^{n} x_i\|$ |
| Type III | Soft constraints | penalize $\frac{\partial Q}{\partial t}$ when $RH(Q, \cdots) > 0.60$ | grad. reg. $\|\mathbf{1}[h(x_i) > \beta]\frac{\partial x_j(x_i, \cdots)}{\partial x_i}\|$ |

The variables $x$ will usually be outputs of the NN but could also be inputs. Type I constraints stem from the fact that
physical variables may be bounded in some way either by definition, a static threshold, or by another variable. For example, precipitation rate and shortwave radiation flux cannot be negative. Likewise, no component of short-wave radiation can exceed the incoming solar flux (given that plane-parallel radiation is assumed in CAM). The constraint, in this case, limits these variables to the space of solutions $\{x | x \in R, x >= 0\}$ and $\{x | x \in R, x >= \text{SOLIN}\}$, where SOLIN is solar insulation variable in CAM. We have found that ML approaches struggle to consistently obey this type of constraint to the required accuracy,
especially since even small violations are unphysical and can cause problems elsewhere in the model, hence we enforce the positivity constraint by applying the rectifier function to the output variables: $max(x, 0)$. This is a type of inductive biasing Karniadakis et al. (2021). Type II or "redundancy" constraints formalize a relation among variables, which generally means that the target solution exists on a manifold inside the unconstrained set of target solutions, i.e., there is physical redundancy in the output variables. For example, the sum of direct and diffuse shortwave radiation at the surface must be equal to the total
radiation at the surface. Type II constraints, therefore, take the form $x_j = \sum_{i=1}^{n} x_i$ where $x_j$ is a surrogate output that consists of $n$ components $x_i$ represented also by other surrogate outputs. To improve overall model learning we incorporate these physically-dictated constraints by converting them to a residual loss term in the objective: $||x_j - \sum_{i=1}^{n} x_i||_2$ such that when the constraint is satisfied the residual will be zero. Note that such constraints could also be nonlinear. Type III or "soft" constraints penalises solutions that disobey a desired physical property expressed via functions of the output variables. For example, we
expect that the sum of all tendencies of water vapor in a column will balance those of condensed water plus precipitation, but





this relationship may not be exact due to small terms in the conservation equation (such as storage of condensed water) that are not known or available. Type II and III constraints are examples of learning biasing Karniadakis et al. (2021). As listed in Table 2, we implement a soft constraint that penalizes increasing moistening tendencies when relative humidity is above a specified threshold with gradient regularization term $|\mathbf{1}[h(x_i) > \beta]\frac{\partial x_j(x_i, \cdots)}{\partial x_i}|$. This term will penalize positive slopes, i.e., gradients of the output variable $x_j$ with respect to input variable $x_i$ but only when $h(x_i) > \beta$. In this example, $x_j$ is the moistening tendency, $x_i$ is the input humidity, and $h()$ is the relative humidity function.

Naively, these constraints could be applied at runtime in the GCM's integration. While necessary, we found evidence to suggest that this approach is insufficient, since it does not address the joint state of all the predicted climate variables. For example, one might coerce a precipitation variable to be non-negative at runtime. However, suppose the learning process had predicted negative precipitation. In that case, this may imply non-physical predictions of other variables that are empirically related to precipitation even if no other variables are violated. To ameliorate this issue, we bring Type I-III constraints into the learning stage, presenting them as additional terms in the loss function. Type I constraints are also applied at run-time since learning does not completely eliminate violations and even small type-I violations (for example negative solar fluxes) can crash the land model.

### 4.3 The skill of the hybrid CAM-ML model

We evaluate the hybrid CAM-ML model by comparing it with standard CAM, whose physics the surrogate emulates. The free-running hybrid model produces zonal-mean moisture and temperature tendencies that closely resemble those of the original model (Fig. 6). The ITCZ is slightly too narrow and there is a double-ITCZ bias in the hybrid model and too much heating near the tropopause, but the differences overall appear modest. The hybrid model also shows similar temporal variations of tropical variability and waves when comparing them to the original CAM model (Fig. 7). This is a stronger test of the surrogate model since the character of tropical waves is sensitive to the behavior of moist physics (Majda and Khouider, 2002; Kelly et al., 2017). Note that the initial behavior of the hybrid model matches that obtained with CAM parameterizations, diverging later as expected but retaining a similar character. Despite spatiotemporal pattern similarities, the hybrid model exhibits larger relative humidity extremes (Fig. 7). It is not clear why these extremes are permitted by the surrogate model, although earlier versions of the model that lacked the RH-sensitivity regularization (Section 4.2) showed a more severe manifestation of the problem suggesting that even with regularization the surrogate model is insufficiently quick to condense water above saturation compared to the CAM physics. This is not too surprising as the parameterized physics is extremely nonlinear and we have not explicitly coded the Clausius-Clapeyron relation into the DNN, so it must learn the point at which condensation (rapidly) begins.

Further insights into the behavior of the hybrid model are gained using the third mode of operation of TorchClim, where the ML model is called alongside the original parametrization of CAM. Here we call the ML model and output its response, without assimilating it into the CAM workflow. Figs. 8-9 compare specific humidity and temperature tendencies of the original CAM parametrization to the ones from the ML model at various vertical levels, between $10°$ North and $10°$ South from the equator. A perfect ML model would follow the 1:1 line. Both specific humidity and temperature tendencies exhibit a line







**Figure 6.** Zonal mean specific humidity and temperature tendencies at the fifth month after the start of the simulation. Showing the original parametrized CAM (left) and the hybrid CAM-ML (right).

of best fit that is smaller than one. The ML model generally predicts smaller positive specific humidity tendencies (Fig. 8). Interestingly, for both quantities, the lowest skill is found in the mid-troposphere between hybrid model levels 510 and 696. In these levels, the ML model learns spurious features of specific humidity tendencies that do not exist in the original model. These discrepancies appear at vertical levels where shallow convection is active and are in line with Mooers et al. (2021), who



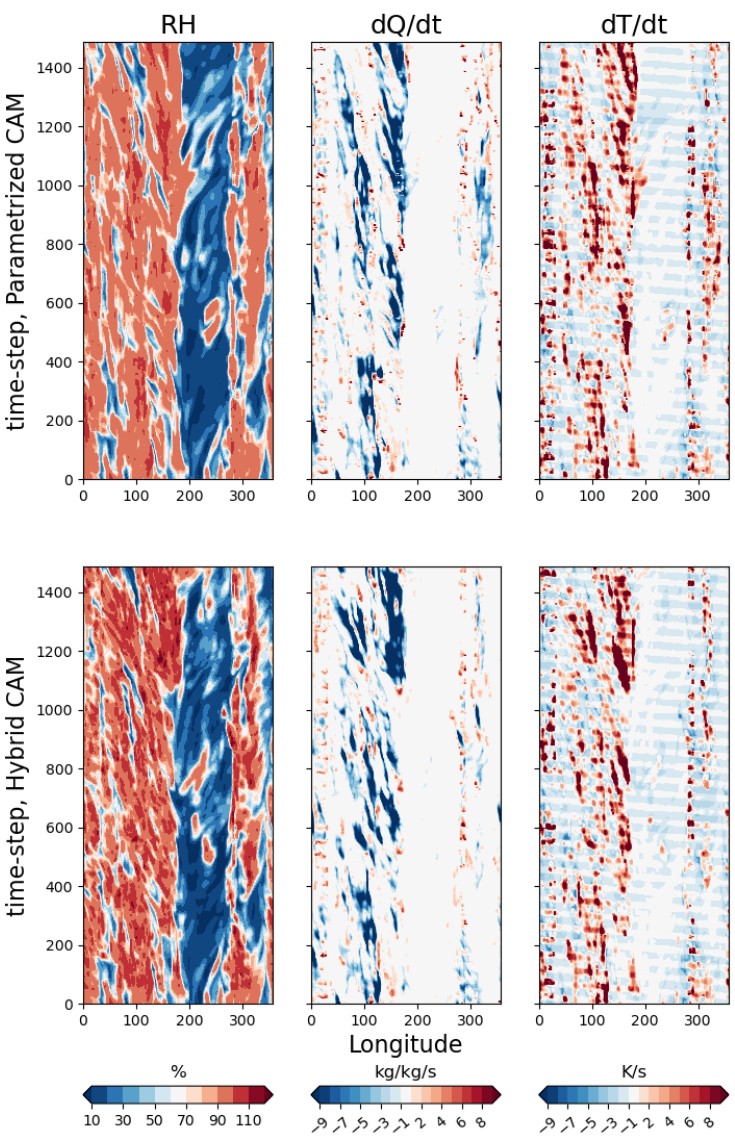

**Figure 7.** Predicted values vs. longitude and time during the fifth month after the start of the simulation at hybrid model level 233 at the equator (approximately 233hPa). Showing the original parametrized CAM (top) and the hybrid CAM-ML (bottom), and left to right relative humidity, specific humidity tendencies, and temperature tendencies. The y-axis denotes the model physics time-step (which is 30 minutes).

found that their deterministic DNN had less skill in regions with fast stochastic convective activity. However, this issue could
also be attributed to the fact that cloud liquid and ice are not treated by the current surrogate DNN model.



Although these biases could be discovered offline during training, we found that the ability of TorchClim to run the original and hybrid CAM models side-by-side enabled us to quickly diagnose errors in hybrid simulations, particularly where there may not be analogs in the training dataset.

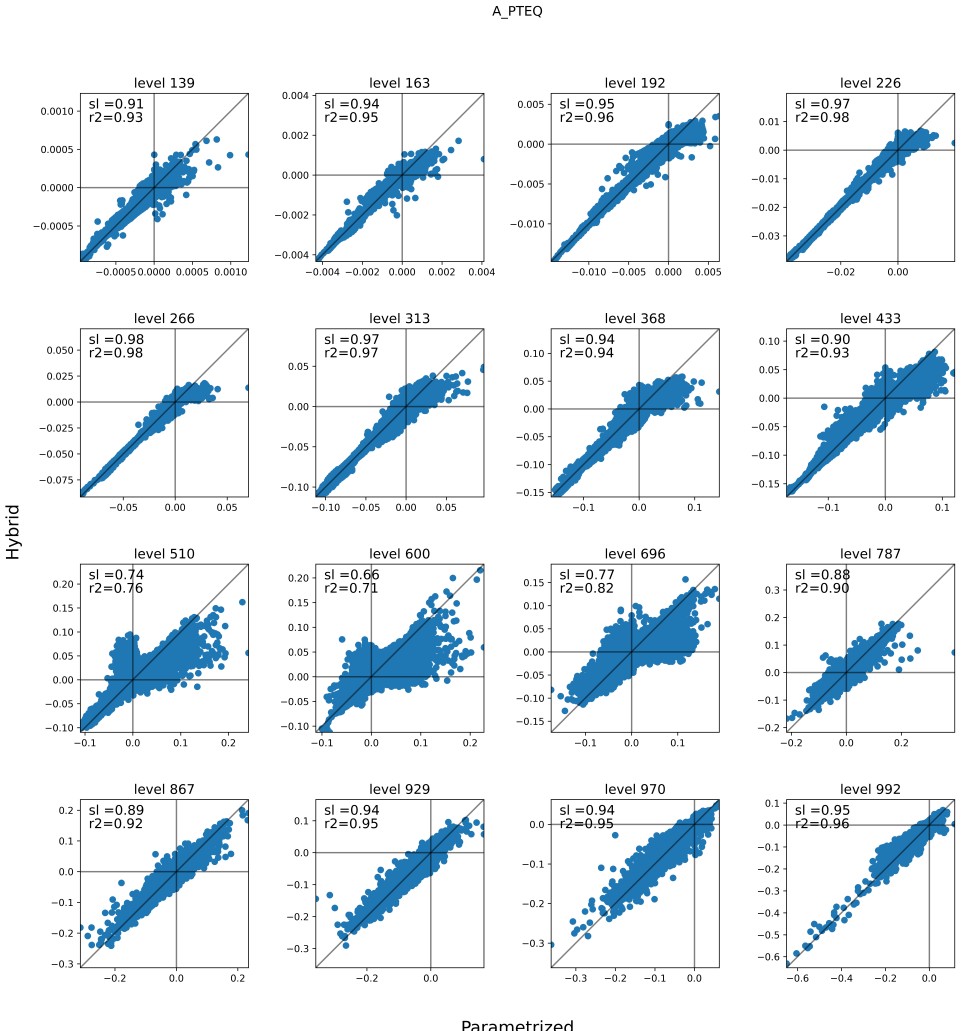

**Figure 8.** Specific humidity tendencies (kg/kg/day * 10), of the original cam parametrization (x-axis) and the hybrid CAM-ML parametrization (y-axis) for various vertical levels at day 10 of the run. Calls to the ML model were done from within the integration loop of the original CAM parametrizations. Black lines denote the origins and the 1:1 line. The hybrid model level is denoted in the title, while the slope (sl) and r-squared (r2) are printed inside each panel.



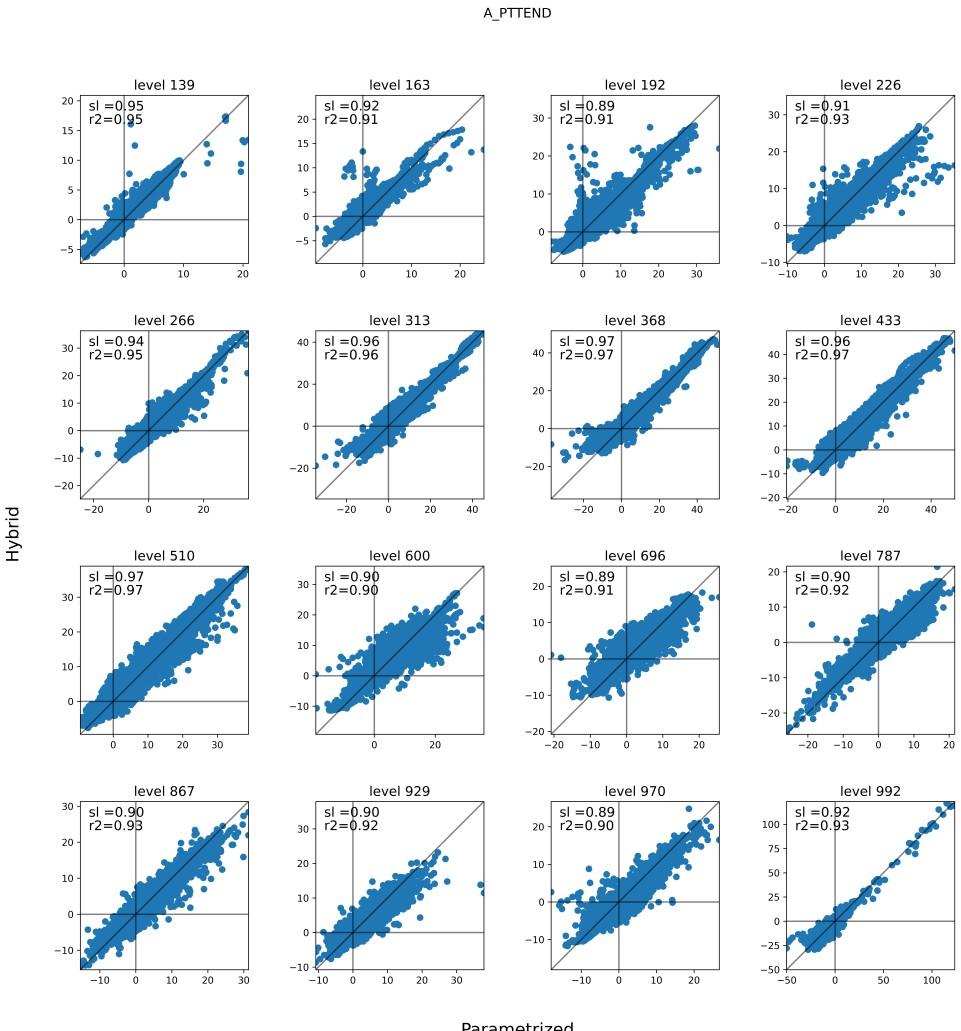

**Figure 9.** As in Fig. 8 for temperature tendencies (K/day).

## 4.4 Computational performance

The evaluation of computing speed is undertaken on the normal queue of the Gadi infrastructure in the National Computational Infrastructure (NCI), supported by the Australian Government. The normal queue nodes are based on 2 x 24-core Intel Xeon Platinum 8274 (Cascade Lake) 3.2 GHz CPUs, with 192GB RAM, 2 CPU sockets, each with 2 NUMA nodes, and no hyper-threading. Noticeably MPI ranks are bound to the core, with each MPI rank hosting an instance of the DNN model.

For the sake of comparison, both original and hybrid CAM configurations were run for a year with minimal monthly-mean
output variables so that a relatively large fraction of the overall CPU would be spent on computation. Each run used three nodes



on the normal queue (144 CPU cores) and 64GB RAM. This configuration matched the number of longitudes of our AMIP scenario spatial discretization. The hybrid and standard configurations of the GCM required similar resources, unlike previous implementations of which we know the DNN implementation is much slower. The results of this experiment are detailed in Table 3. The hybrid model added 20 percent to the wall-time of CAM, reducing its performance by 8 modeled years per wall-

time day. The total CSEM performance degraded by 10 percent, which amounted to 4 modeled years per wall-time day. The addition of ML/AI model added about 5 percent (2GB) to the overall memory requirements.

Since our setup is similar to Wang et al. (2022), it is convenient to compare the performance of these frameworks. In their work, Wang et al. (2022) used a similar 1.9 x 2.5 degrees grid resolution and 30 instead of 26 vertical levels. Their run used CAM5, which is about four times slower than CAM4. Noting uncertainties regarding the computational complexities

of the DNNs of each implementation, their configuration was able to produce 10 modeled years per wall time day, which is comparable to the performance that our framework offers. However, their approach used 25% more CPU cores, Intel's Math Kernel Library optimization, and additional dedicated hardware (GPU and storage). We also note that our results were achieved before any optimization (such as vectorizing the calls per geographical tile, pre-allocating of memory buffers, using GPUs, etc). In developing future versions of TorchClim, we anticipate vectorizing the calls to the surrogate DNN model, pre-allocating and

re-using data structures, and considering automatically offloading to a GPU when available. It is expected that, despite each MPI rank hosting a single thread, the performance improvements in vectorization and pre-allocation will be significant (even without GPUs or multi-threading). For example, assuming that vectorization will match the spatial tile (chunk in CAM's terminology), the reduction in heap memory allocations calls will be of the order $O(|tile|)$, where $|tile|$ denotes the number of atmospheric columns of the tile (thus being more important for larger tiles in terms of the number of grid locations per tile).

**Table 3.** Run-time performance for CESM total run-time and the atmospheric component (CAM) only.

| Criteria | Original physics | Hybrid reference implementation |
|---|---|---|
| Total duration (seconds) | 2051.411 | 2265.122 |
| CAM duration (seconds) | 1767.797 | 2132.996 |
| Total modeled years/wall-time day | 42.12 | 38.14 |
| CAM modeled years/wall-time day | 48.87 | 40.51 |
| Memory used (GB) | 42.94 | 45.16 |

**5  Gaps and Extensions**

**ONNX support**

The current version of TorchClim is able to import surrogate models that were saved through Pytorch's scripting infrastructure. A natural extension of that is to allow the use of the ONNX interface to imports. With the maturity of ONNX, this would improve performance while allowing to use ML/AI framework beyond Pytorch.





**Stateful ML/AI models**

A stateful AI/ML model preserves an internal state between one call to another. For example, in the case of agent-based algorithms such as reinforcement learning. In the case of GCMs, a specific instance of surrogate model, compute node and GPU may be associated with an MPI rank. However, a standing issue with MPI applications is the need to optimize their performance on a given infrastructure. In the age of non-uniform-memory architectures (NUMA), this tends to be achieved via binding (affinity) instructions that communicate with the underlying infrastructure the required binding of MPI ranks. For example, many applications require binding to cores, memory, or cache to achieve the best performance. The case of the stateful ML/AI model brings in an additional challenge. In this case, an instance of the ML/AI model holds an internal state that is bound to a given geographical location. This has two consequences:

1. A geographical location in the GCM needs to be mapped to an instance of the ML/AI model, which is mapped to a specific thread.

2. If the ML/AI uses GPUs at the GCM run-time, the instance of the ML/AI model (and state) are bound to a given GPU.

The consequence of this is the binding of a geographical location or tile to a resource (thread and GPU). An implementation of this feature requires TorchClim to retain a mapping between a GPU and a geographical location, and for the GCM to bind the geographical location (tile) to a CPU. Since the latter requirement is GCM-specific, its implementation of these is out of scope in the current reference implementation.

**Vectorization and preallocation**

The current implementation is shipped without optimization. Despite that, our experiments with the reference implementation into CESM/CAM exhibit good performance. Beyond that, the framework is desired to automate the vectorization of calls to underlying surrogates. In addition, the framework should preallocate all the buffers required to interact with LibTorch. These improvements are expected to have a significant effect on performance even without the use of GPUs.

**Multiple surrogate models**

The reference implementation does not support multiple surrogate models within the same run. However, the extension of the TorchClim framework to support multiple surrogate models is straightforward.

## 6 Conclusions

This work presents a new framework, TorchClim, that facilitates the introduction of ML/AI-based models into GCMs. It focuses on offering a robust and scalable way to introduce ML/AI surrogate models into GCMs, addressing a key gap in current practices. The framework combines ease of use and flexibility with computational performance in a "plug-and-play" like fashion. It can be loaded and used from any component in the GCM and expose itself as any other parametrization. It can





support any number of surrogate models simultaneously, while surrogate models can be loaded without the need to recompile
the framework. The user can configure the framework to run surrogate models on CPU and GPU configurations (further list of
requirements and features can be found in Section 2.1). This is offered as a community-based open-source project (see code
availability below).

  Though the framework may have a wider range of applications, it aims to offer a data-driven approach to physics parametriza-
tions. To this end, we implement a proof-of-concept into CAM physics, replacing parametrization of moist and radiative
parametrization with a call to TorchClim. We test this by creating a surrogate model that was learned using CAM data. In
doing so, we found the need to assist the learning process in learning stable, physically motivated surrogate models. This was
done by introducing constraints as added loss terms during the training stage. These were generalized into a set of guidelines
that could be reused for other learning tasks. Following this approach, the surrogate model performed well compared to other
models in the literature in a scenario that included interactive land surface. This test was executed using the same hardware
as the original CAM run that produced the training data. It showed similar computational performance between the TP and
surrogate model. Noticeably, it did so without specific optimizations such as vectorization and the use of additional dedicated
hardware.

  We anticipate that the flexibility and speed offered by this framework will help unlock the full potential of AI/ML in ad-
vancing climate modelling, by allowing rapid testing of many candidate physics emulators. Future extensions should consider
training using other data sources in order to improve on existing traditional parametrizations. Our tests with CESM/CAM sug-
gest that the incorporation of TorchClim to other parameterizations, later versions of CESM, and even other GCMs is relatively
straightforward. Our hope is that such a framework will pave the way for the introduction of ML/AI surrogate models into
GCMs.

*Code availability.*  The TorchClim framework is available at https://doi.org/10.5281/zenodo.8390519 (Fuchs et al., 2023a, current version).
User and installation manuals can be found in the README.md file of the project. Further development effort of TorchClim is tracked at
https://github.com/dudek313/torchclim. Scripts and data for the figures in this work are available from the corresponding authors.

*Author contributions.*  All authors contributed to the development of ideas and writing of the manuscript. SS designed the initial experiments
with contributions from all the authors. SS, KT, AP, and DF developed the final format of the TorchClim framework and its reference
implementation. KT implemented and tuned the DNN, and DF and AP ran the simulations that produced the inputs for the DNN, as well as
the test runs with the surrogate model.

*Competing interests.*  The contact author has declared that none of the authors has any competing interests.





*Acknowledgements.* This work was partly funded by the DARPA ACTP Program. The computational support and infrastructure were provided by NCI and CMS team at CCRC/UNSW.



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
