# Peer review of "TorchClim v1.0: A deep-learning plugin for climate model physics"

_EGUsphere, 2023_

## Author Response (AR1)

**Reviewer 1 comments and authors' replies**

The manuscript "TorchClim v1.0: A deep-learning framework for climate model physics" by David Fuchs, Steven C. Sherwood, Abhnil Prasad, Kirill Trapeznikov, and Jim Gimlett describes an application of a machine learning (ML) approach within the CAM AGCM. The aim of this study is to introduce an effective framework that can be adapted to implement ML approaches in climate models development. The sudy has a technical aspect and a climate model related aspect. My background is in climate dynamics and modelling. I cannot really comment on the computer technology aspects. The study addresses an important aspect in climate model development: how to include powerful ML approaches into model development. The manuscript should be considered for publication, but there are a number of aspects the authors should consider before publication.

Reply:

We thank the reviewer for taking the time to review our work. We made every effort to make changes that would make our work more accessible to the wider climate community, removing software-eng. jargon, while providing sufficient information to the primary target audience - those with relevant skills, to implement the plugin into GCMs.

Main comments:

(*) Outcome: To summarise the manuscript: The study implements the TorchClim code of ML to improve one specific parameterisation in an AGCM. It is designed to be "plug-and-play", but to do so they actually have to heavily optimize the ML algorithm for the specific problem/model, which appears to be not "plug-and-play" at all.

Reply:

We added further information in the requirements (Section 2.1 bullet point 3) explaining the scope of the plug-and-play requirement. Briefly, it consists of the ability to test different ML/AI models minimizing the need to recompile the GCM and the plugin. We also made it clear that this does not include internal changes to the GCM.

…As an outcome the model is not as good in presenting the climate state and it is numerically slower than the original model. Given this outcome you would wonder: why would anyone want to use TorchClim? The authors seem to be presenting a failed approach. While this is still helpful for the community, it is unlikely to motivate the readers to use this approach. The authors need to think about how this study can be more helpful for the reader.

Reply:

We added to the introduction the various motivations for using our approach ("There are multiple motivations for developing hybrid models…"), and have also reworded text elsewhere and especially in the Conclusions section to clarify the value of what we have done.

As noted in these revisions, the current release of TorchClim focused on functionality rather than the performance measures noted by the reviewer. We are currently working on a vectorized version where the entire MPI tile will be fed at once to the plugin (see Section 5). The main premise in the current approach is that a DNN of similar complexity but trained on

other sources such as observational datasets and high-dimensional data, could improve the model without exhibiting the increase in computational overhead associated with more sophisticated parameterizations or higher resolutions. For example, Wang et al. (2022) learned a DNN from a super-parametrized CAM (SPCAM) and used it in a lower resolution CAM. The hybrid model displayed properties of the SPCAM with computational performance closer to the low-resolution CAM. Note that our approach seems to offer a similar performance compared to Wang et al. even without added resources.

(*) Technical aspects: The manuscript is primarily written for a computer science audience, and not as interesting for the climate model researchers. Given the journal being read mostly by climate researchers, I would recommend to strengthen the climate modelling aspects and reduce the technical aspects, as they appear less important. However, I may have missed the importance of the technical aspects. One way to do this may be to put the simulation results first and explain some of the technical aspect later or in an appendix.

Reply:

At the time of writing there are more than 800 views and downloads of the manuscript, a few direct inquiries, and a citation. Seemingly the community is interested. Note that the climate science community is diverse, with a portion of it having, by necessity, substantial software engineering skills.

However, we recognize the reviewer's point that the original manuscript could be better written and have now made extensive revisions in several sections of the paper (especially 2.1, 2.2, and 3 but also the introduction and conclusions) to clarify the approach.

Other changes to the manuscript:

- Replaced Fig. 1 with a figure that provides an entry level overview of the plugin.
- Where possible, tuned down the engineering jargon (without compromising the ability of the target audience to implement the plugin into a GCM).

(*) Insufficient results section (section 4): the discussion of the application results (figs. 6-9) is too short. It should be explained what experiments have been done, how long have they been running, what time interval after initialisation have been analysed. An Important aspects of an ML parameterisation is stability. So, running longer (several years) simulations without running into numerical instabilities, or unphysical states would be important to illustrate.

Reply:

Added further information to Section 4.3:

"Here we present results from a hybrid CAM-ML run that starts from the same starting point that was used for extract the training data. That is CESM AMIP scenario for CAM4 with default configuration parameters. The hybrid CAM-ML model was able to run for six months before exhibiting numerical instabilities. Here we present results for the fifth month into the run, before the instabilities became apparent. During this month the hybrid model produces…"

Other comment (in order as they appear in the text):
* * *
line 14 "global circulation and climate models (GCMs)": GCM = Genreral Circulation Model

**Changed accordingly.**
* * *
line 26 " For example, Fuchs et al. (2023) showed ...": A selfcitation is not a convincing argument. Can the authors find independent literature support?

**Added references.**
* * *
line 31-32 "Yet in some cases, the sheer number of parametrizations and versions of parametrizations in current GCMs could point to a fundamental problem with TP": Unclear what the authors want to say here. Why/How would "the sheer number of parametrizations" point to a fundamental problem with TP?

**Rephrased to:**

"The number and variety of parametrizations in current GCMs arguably points to a lack of consensus on appropriate conceptual models of small-scale processes and how they work."
* * *
line 46 "the curse of dimensionality": What is this?

**Removed from the text to make the manuscript more accessible to the general climate community.**
* * *
line 46-47 "... the increase in the computational complexity of GCMs was met by ... . The increase in computational complexity of GCMs was met by ... ": Repetition.

**Fixed, thanks!**
* * *
line 57 "One way of doing this is to use observations to tune existing or new

parametrizations": One would hope that observations are always part of parametrizations. Not sure what the authors want to point out here.

**Rephrased to:**

"One way of doing this is to use ML to adjust physics parameters at run time to steer the model toward observed states."
* * *
Fig. 1: The logic and meaning of this diagram is unclear. It is also unclear why a climate modeller would or need to care about this.

**Removed the figure.**

Figure 1 is now a simpler one offering a broad view of the solution that should be more accessible to the general audience.
* * *
CPUs vs. GPUs: Why does a climate researcher need to care about these technical aspects? If this is mentioned in the manuscript, it needs a bit more explanation for non-computer science audience.

section 2.3 "Parallelization framework": Why is this important? CPU speed?

Naively, a climate modeller may think the sole purpose of the ML approach is to compute the parameters of a complex function and then the complex function is replacing the TP equation. Then why would I need this whole infrastructure here? It seems to be overkill.

**Reply:**

Following the reviewer's comments, we changed section 2.3 making it more accessible to the general audience and added more background explaining the importance of this to the introduction.

**section 2.2 and Fig. 2: It may help to start with the physics part and then go to the LibTorch block. At the end the authors write for climate researchers not for computer scientists. Or maybe both, but it still would help the climate researchers to get into this.**

**Reply:**

This section is modified following the reviewer's suggestion. Figure 2 (Figure 1 in the new manuscript) provides a simpler view from the GCM perspective, offering an entry-level view of the overall solution.
* * *
line 207 "... three categories": What is the important difference between the categories 2 and

3? They seem to be similar. Whether it is a dynamical or other kind of model does not appear to make a difference.

**Reply:**

We replaced the three categories with Table 2 and expanded on it to better suit our aim. We explicitly state that the categories are not mutually exclusive. The aim is to help the user identify which variables need to be included in the outputs of the surrogate model.
* * *
Table 1 and other sections: Lots of variable acronyms are presented but not explained (.e.g, what is FLNS?).

**Added reference to CESM/CAM 1.0 documentation.**
* * *
line  229-230 "We note that a good starting point for training a surrogate model ... is simply the TP":

Really? Evidence for this? It could limit the design of the new model to be arbitrarily similar to a bad TP model.

**Added further details.**

"A good starting point for training a surrogate model is simply to use the TP of the host GCM itself. In the reference implementation, this is the combined effect of the moist-physics and radiation parametrizations of CAM. This approach has the benefit that it allowed us to benchmark a surrogate model both computationally and scientifically against an ideal synthetic dataset. It can also serve as a starting point for further training from other datasets that have missing or insufficient data, as is frequently the case with climate data. User could mitigate learning biases in this using the TP as a starting point in different ways. For example, by using an ensemble of TPs, potentially from different GCMs, or increasing the learning rate when training with subsequent data sources."
* * *
line 232-233 " .. the first is inserted in the TP before ... the second is inserted after that point.": Why? What is the purpose?

**Reply:**

This is a convenient tool to get training data for the surrogate model. As seen in Fig. 4 (Fig. 3 in the new manuscript), the first insertion point records input state at the desired point of insertion of the surrogate, while the second point of insertion records outputs.

**Added to the text:**

"These produce a dataset of inputs and labels required for supervised learning."
* * *
Fig. 5: Not clear what this diagram is explaining.

**Rephrased the figure caption:**

"The process of recording the state and tendencies in a TP case that will be used in the ML/AI learning process…."

To:

"The process of producing inputs and labels for supervised learning from the TP parametrization…."
* * *
line 239 "these data": Which data? Unclear.

**Removed text and changed the opening sentence.**
* * *
Section 4.1, DNN model development: The description here sounds highly specific for the particular problem, with statements like "tested dozens of versions" you get the impression that this is not "plug-and-play". As the aim of the study seem to be to introduce a general approach, it would be helpful to see how this experience can be used to explain a general application. It does not seem straight forward.

**Reply:**

Testing many surrogate models (quickly) is precisely what we wanted to achieve from the plugin. The term plug-and-play refers to this property. It comes to support a period of exploration of different surrogate models. The description is specific to modelling moist convection and radiation parametrizations which is specific to the reference implementation.

The text modifications noted earlier should clarify what is meant by this claim and why it is useful.
* * *
line 265 "MSE": not explained.

**Fixed.**
* * *
explanation of lost function, eq. in line 270: The lost function is not well explained. A number of terms play into this and later in the text more terms are mentioned and it is unclear how they relate to the equation in line 270.

Line 270 defines the loss function (rather than "lost").

**Fixed.**
* * *
line 266 "L2 regularization": What is this?

**Reply:**

See similar use of this common terminology in the climate context in JAMES: https://agupubs.onlinelibrary.wiley.com/doi/full/10.1029/2022MS003219. To keep the manuscript to a manageable length and readability, we do not attempt to explain the meaning of every methodological detail in the ML implementation for the benefit of non-ML readers, but clearly must provide these details for transparency and repeatability. Interested readers who don't already know them can quickly Google terms like this. We have tried to revise the text to ensure that the basic points are clear even to readers who aren't up on the technical ML side (and likewise for those not familiar with the construction of climate models).
* * *
line 296 "loss term": How does this related to eq. in line 270

**Reply:**

The loss term equation in line 270 was changed to match the terminology in Table 2. Thanks!
* * *
line 301 "... examples of learning biasing Karniadakis et al. (2021)": unclear. Maybe: "examples of learning biasing following Karniadakis et al. (2021)"?

**Fixed.**

Thanks for the constructive comments!

---------- end -----------

**Reviewer 2 comments and authors' replies**

Reviewer:

A major challenge of hyrbid ML climate modeling is ensuring that the simulations are numerically stable over long roll-outs and the learned parameterization does not diverge. It would be nice to see some high-level discussion about how TorchClim may be useful for monitoring/addressing this in Sec. 5.

Added Section 4.5 (How might TorchClim help with stability issues)

Added to Section 3.1 (Alternate physics workflows):

"…it may be desired to run multiple models side-by-side or as a replacement for different physics parameterizations. This functionality could also be used to diagnose instabilities in the surrogate model or do online corrections… For example, by comparing the outputs from the surrogate model with the TP."

Added to Section 3.2 (Assisting the learning phase):

"This functionality could also be used to study instabilities during online runs of the GCM with the surrogate model. In our reference implementation, this can be done by extracting samples before and after the call to the surrogate model and tracking offline the locations where the surrogate model diverges from the TP."

Reviewer:

Can this framework accommodate ML/AI surrogates trained on higher resolution models or observations? How will Fig. 5 be modified in that case?

Reply:

We have added to the revised Conclusions an additional note that high-resolution simulations are a clear next step for generating training data.

Reviewer:

Typically ML models are evaluated using hold-out validation and test data sets. While the authors indicate that they held out one year of data for testing (L250), it's not clear if Fig. 6 and 7, and the relevant discussion, were derived using test data. If they were, a clarification in the text would suffice; if not, I would recommend including additional discussion of similar plots using test data.

Reply:

These figures are based on a GCM run with the surrogate model plugged in.

Added a clarification to the captions, emphasizing that this is not an evaluation of the ML/AI model but rather an online run of the GCM coupled with the ML/AI surrogate ("…, where CAM is coupled with the ML/AI model using TorchClim"). Figures 7 and 8 in the new manuscript show clear discrepancies between the GCM run with the TP (the truth) and the online run with the hybrid model.

Reviewer:

It would also be useful to evaluate the difference in output between the TP and ML surrogate models spatially for 1-2 pressure levels as shown in Fig. 3 of https://arxiv.org/abs/2309.10231

Reply:

As said, our focus is the plugin and its recommended mode of use. We will follow up on the reviewer's suggestion in follow-up studies that the authors are conducting using TorchClim.

Reviewer:

Is there any particular reason why the hybrid run is slower than the TP run (Table 3)? Does it not defeat the purpose of using a ML surrogate for TP? Especially for the surrogate model presented here and not in the cases where the parameterization is "learned" from high-resolution simulations or observations. Relatedly, are there any significant computational costs to using a PyTorch model with a Fortran interface?

Reply

- In this release of the plugin we focused on functionality rather than performance. We revised the text to explain better why we focus on this. The next version of TorchClim will include vectorization of the entire MPI tile (see Section 5) which will improve performance and open several other use cases. Note also that we did not compile LibTorch for the target CPU which could also slow things.

- CAM4 is a fast model compared to later versions of CAM, so we think that roughly equalling the speed of CAM4 is a positive result, given that previous efforts have been significantly more compute-intensive than their parent GCM (e.g. by using a GPU). We have clarified the text on this point. Our performance would likely be superior to CAM5 for example, if emulating CAM5. Ultimately the TorchClim plugin is quite lean and bounded, so the computational requirements depend on the NN. What is important is the asymptotes of time/space complexities between how future versions of GCMs with TPs will scale compared

to hybrid models. The assumption is that NN-based surrogate models will scale better and there are some positive results to support that (e.g. Wang et al., 2022). This assumption was added to the manuscript ("…The main premise in this case is that the increase in computational complexity…"). Our surrogate DNN is also quite complicated and lacks optimization, so we think that it is comparable with DNNs that will include additional datasets.

Minor comments:

- L151: "a Fortran interface in the reference implementation" -- "in" should be replaced by "is."
  Rephrased to:
  "Offers a workflow and supporting tools to boost the learning process of ML/AI models. Since most GCMs are written in Fortran, offer a Fortran in addition to c/C++ interface in the reference implementation;"

  Fixed.

- L156-158: Rephrase "As such … byproducts." -- as written, the goal of these statements is unclear; these may be removed as well since the bullet points adequately outline TorchClim's contributions

  Removed.

- L263: Use "renormalization/rescaling" instead of "denormalization" since the latter has a specific meaning in database management

  Fixed.

- L272: The number of training epochs seems low; what are the results when the DNN is trained for 500 and 1000 epochs, or 100 epochs with smaller batch size?

  Added:

  We determined 100 epochs to be a good compromise between model accuracy and training time on a separate validation set. Training beyond 100 epochs resulted only in marginal gains.

**Reviewer 3 comments and authors' replies**

General comments:

This manuscript describes a framework for coupling Pytorch models into existing compiled models, and is employed within CAM to use a ML surrogate for an existing physics parameterization. Generally speaking, a framework simplifying the deployment of ML/AI models using Pytorch into models such as CAM is a welcome contribution. However, as written the motivation for this framework and how it is meant to facilitate use of ML/AI models is unclear. I would suggest major revisions to the manuscript to clarify what is being presented, how it is meant to be used, and why it is useful.

**Reply:**

We thank the reviewer for the time, important insights, and the technical perspective. We have made substantial revisions to the manuscript to address this concern which was also shared by other reviewers.

It is somewhat unclear from the manuscript what the TorchClim framework _is_. From what I have gathered, it consists of a wrapper over the torchscript model allowing it to be called from Fortran, and a second wrapper layer responsible for packing data into a single array so the torchscript model can be called. This second wapper layer is described as being model agnostic, but hard-codes a set of variables specific to the ML component and the host model. The synthesis of this appears to be "we call a torchscript model directly using the host model data, treating it as any other model component instead of through an intermediary layer".

This is not really a framework, but not everything needs to be a framework. It can be enough to show and describe an approach to wrapping ML models, without calling it a framework. Regardless, it must be made much more clear what this contribution is, how it could be used by the reader, and what benefit it would bring if they use it.

**Changed:**

- Changed the term "framework" with "plugin". The abstract was also modified accordingly.
- Added a clarification regarding the mode of use of the plugin and elaborated on the fortran layers in the plugin and how the user might benefit from them (Section 2.2 "The first release of the plugin is shipped with a Fortran layer that packs variables...").

Note that we use Tochscript just as a starting point because options like ONYX and TorchServe seem not fully mature. The right way to look at it is that we offer an interface that hides this implementation detail from the GCM. Users should be able to benefit from those features when they are ready without needing to change the implementation on the GCM-side.

It is also a significant issue that the presented AI/ML model is slower and worse than the existing physics scheme. Given the emphasis on performance, it should be possible to produce a physics scheme with improved performance, though skill is not a priority in this context. If the authors are not able to achieve better performance, this result should be emphasized and its implications explained to the reader.

**Changed:**

- Added a list of other goals, aside from computational performance (see introduction "There are multiple motivations for developing hybrid models…")
- Added a statement regarding the assumptions behind the use of ML/AI as surrogate modes (see introduction "The main premise in this case is that the increase in computational complexity…")

As explained in Section 5, the first release of TorchClim comes without optimizations. The fact that it produces comparable computational performance to the TP is great considering that we use exactly the same resources as the TP without optimisations and a relatively small number of cores (adding a GPU or an additional process pool would make it an unfair comparison). The intent is to offer the user, later, other out-of-core options without forcing it on him architecturally (as was done in WRF-ML).

Computationally, the issue is mostly the underlying ML/AI model which is outside the scope of our work. The premise in going down the ML/AI path, is that ML/AI surrogate models will exhibit better computational scaling compared to traditional approaches (time complexity, development cycle, etc). This is something that needs to be demonstrated on production workloads over time. We have some encouraging results, for example, Wang et al. (2022) incorporate data from a super-parametrized CAM into a low-resolution CAM, showing similar performance to the low-resolution CAM. Note that they used a GPU so, unlike us, this is not a fair comparison.

Specific comments:

The manuscript focuses heavily on this concept of a "bridge pattern" used by existing frameworks, described as where the global model state is gathered and then transferred to GPU which re-defines the parallelism scheme. Particular focus is put on these frameworks requiring transfer to GPU, which can be a performance bottleneck, emphasizing that their framework supports CPU operation using the existing MPI domain decomposition. However, I am not aware of an existing framework (the two cited included) which requires the use of a GPU, as it is typical for ML codes to work on either CPU or GPU. If the emphasis is on avoiding a global gather, particular focus should be placed on this, but I am quite surprised to hear that existing frameworks place this kind of requirement. The ones I've seen, including my reading of WRF-ML cited by the authors, incorporate the ML model directly into their parallelism scheme rather than define a new one solely for the ML component. Even if this were the case, it is unclear why running the ML on CPU would be an optimization need worth developing an entire framework - typically the optimization requirement model developers have is to run their model components on GPU, not take them off of GPU.

**Changed:**

- Removed our definition and use of the term bridge-pattern and discuss each of the items in our definition of "bridge-pattern separately.
- Removed Fig. 1 and 2 and replaced them with a figure that is accessible to the scientist and replaced the term bridge pattern with more accurate terms (e.g. network, out-of-core, etc).

None of the implementations of similar tools tie the hands of the user to a given mode of use, including TorchClim. The issue is that most GCMs use CPUs and MPI, so presenting the use of GPUs or introducing a second parallelization framework in the GCM or going out-of-core (where the MPI rank resides) to do calculations should not be the only thing that is tested and offered to the user as best practice. At least not as the only option for inference of small to medium DNNs. All of these add another bottleneck to future scaling, more infrastructure, cost, and potentially non-optimal use of resources. At the least, the computational performance of these is not fairly compared to GCMs with TPs (that do not use ML/AI surrogates). In our work, we start by using the CPU core that is bound to the MPI rank because it is a fair comparison and a fair question to answer. The fact that the use of the CPU shows performance that is not too far from the traditional parametrization, even without optimizations or additional compute, and similar to Wang et al. (2022), suggests that it is an option to consider. Users should be advised not to ignore this option and benchmark before running expensive workloads.

If a goal of the paper is to produce performance improvements, it is critical the authors show results where an inline ML emulator of the physical component is more performant than the existing physics. If the cause of the lower performance in the hybrid case is ML model complexity, a simpler model should be used. If the slowdown is due to the framework, the goal of this paper should instead be to maintain or avoid significantly degrading performance while providing a significant scientific or process benefit.

**Changed:**

- Added to the introduction
  """"The main premise in this case is that the increase in computational complexity of ML/AI models will be significantly smaller than the a TP with similar skill.""""
- Changed the abstract to better reflect the main deliverables of the paper.

- Discussed other the breadth of motivations behind our work (see introduction: "There are multiple motivations for developing hybrid models…")

The slowdown is likely related to the ML/AI model and the fact that the first release of TorchClim does not include vectorization and pre-allocated memory. The first release focused on functionality and the requirements are detailed in Section 2.1. Note that CAM4 is relatively a fast model. We could easily use a more complex and slower version of CAM or more CPUs, but the important issue is how ML/AI models will scale with the addition of new data sources, compared to further improvements in TPs. This is spelled as an assumption in the new manuscript and remains an open question ("The main premise in this case is that the increase in computational complexity…").

In general, the manuscript should focus significantly less on this pattern for performance optimization, and instead emphasize its scientific and workflow benefits.

As a scientific or workflow tool, it should be made much more clear how this framework is meant to be used by a scientist. There was some discussion of saving output data from the host model to use for ML training, it would be good to have more of a sense of what this means

**Reply:**

Following RC1 and RC3's advice we revised the abstract, introduction and Sections 2.2 and 2.3 of the manuscript to make it more accessible to the general scientist. Note that adding the plugin to the GCM will require some technical skills.

Does the framework support ML schemes that communicate in the horizontal, or is it limited to column-local schemes?

**Reply:**

The plugin is not limited to the use case that we present and could be used for the scenario that RC3 describes, but that requires a few more steps in the implementation which go beyond our recommended mode of use. We are interested in an implementation of this scenario that avoids the MPI gather within the GCM and might test implementations that wrap TorchServe. This might be more relevant once we vectorize the calls to the plugin.

Line/section comments:

L55-77: Given the focus on improving existing parameterizations in hybrid modeling, another approach to include is the use of online bias corrections, e.g. Watt-Meyer et al. 2021 https://agupubs.onlinelibrary.wiley.com/doi/full/10.1029/2021GL092555. This uses another model parallelization approach which isn't yet discussed here, giving Python the responsibility for MPI decomposition by wrapping the Fortran model: https://gmd.copernicus.org/articles/14/4401/2021/

**Added Watt-Meyer et al. (2021)**

L119: "Such an approach likely cancels the scalability benefits of GPU, due to the data-transfer overhead over media that is many times slower than CPU access to local memory." This is not necessarily true, it depends on the relative speed of the CPU and GPU implementation and on the amount of data being transferred. Can you produce one or more references to specific cases? An example of a model with dynamics and physics on separate hardware would be relevant.

**Removed text.**

The issues are latency to access the GPU and bandwidth (throughput) in the access parallel case. Larger DNNs could favor inference on the GPU but these could also have larger input/output dimensions and slow the GCM too much for some applications. We note that our reference implementation shows performance that is similar to Wang et al. (2022) without the added resources and without optimizing our code. But this might be an isolated case. Users should be

advised to test their configuration on a case-by-case before running expensive workloads and the underlying plugins should facilitate that.

L122: "Moreover, since most current GCMs are based on CPUs, the GPUs will be idle during much of the GCM execution (and vice versa for CPUs)." This is not necessarily true in a parallel model, and it's not obvious that if it were true one would rather run everything on CPU. If the model coupled to GPU is faster, does a modeler care that the GPU is mostly idle? Some discussion of this (e.g. "when physics parameterizations are processed serially", "leading to increased runtime despite the faster GPU processing speed" (with reference)) would be helpful.

**Reply:**

Section 2.3 was modified following the request of RC1 to make it more accessible to the general community. We adopt the reviewer's (RC3) comment and incorporate it to the modified text adding:

"""This is especially relevant to implementations that access the GPU serially or use a global MPI gather to push data to the GPU."""

In the next release of TorchClim, once we implement the vectorization option (see Section 5), we will be able to test the scaling of this approach and provide more insights comparing to the original (TP) of the GCM.

L117-125: This paragraph seems to imply that existing bridge-pattern implementations do not support running on CPU, but WRF-ML supports CPU inference: "While GPUs are typically more powerful than CPUs, both CPUs and GPUs are supported for inference. CPUs are cheaper than GPUs and more widely available, and using CPUs for inference can also avoid extra costs due to GPU–CPU data transfer." NN-GCM supports CPU models as well.

The conclusion of this section appears to say that because existing coupling frameworks don't allow running the entire model on CPU (not true) which makes them slower (unclear if true) the main objective is to produce a framework that keeps the data on CPU (something which has already been achieved). This is unlikely to be the message you intend to get across.

**Reply:**

The paragraph is reworded to clearly state the issues.

WRF-ML is an interesting solution, and it is true that they compare CPU vs GPU performance, but they force going out-of-core from the CPU-core that hosts the MPI rank of WRF (which was part of our initial self-made definition of the bridge pattern). We also did not find a mention of the overall performance compared to WRF with traditional parametrization and note that their implementation loads a python interpreter which is not great. Perhaps this implementation is needed for WRF? It has the benefit that it isolated the GCM from the memory footprint of the ML/AI model and could help with surrogate models that require synchronization for spatial information beyond what the traditional parametrization has. It would have been interesting to see the fraction of WRF's runtime

that the GPU or CPU process pool are, if at all, idle. My guess is that the user will need to do substantial tuning to balance performance vs idle time of resources.

Introduction in general: The introduction feels a little disjointed. It starts with a (well written) extensive discussion of the current state of machine learning integration into climate models and the scientific difficulties these face in achieving trustworthy, stable, generalizable, skilful predictions. This discussion gives the reader a good sense of the main difficulties preventing these approaches from getting wider adoption. But then the end of the introduction pivots from these scientific difficulties to a performance optimization (keeping everything on CPU), where it's unclear whether that performance optimization is novel or needed. Does having an accessible coupler help more directly with the scientific difficulties discussed earlier?

**Reply:**

Changed the end of the introduction following the reviewer's recommendation (L75 onwards).

I would suggest revising the introduction to better motivate the creation of this framework under scientific grounds, while also describing its computational benefits. It would be great to have more of a sense specifically how this framework will help ML/AI surrogates find their place as drop-in replacements.

L168: Should elaborate "packing and normalizing" if the framework handles normalization.

**Removed text.**

It does not handle normalization.

L181-182: "However, ML/AI frameworks tend to scale better when using GPUs." What does "scale better" mean here?

**Changed to:**

""""We note that most GCMs rely on CPUs to parallelize their execution and scale via MPI to multiple CPUs and compute nodes, while ML/AI frameworks tend to scale using GPUs.""""

L183: "Other approaches mostly relied on the bridge pattern, which generally means a blocking step inside the climate model that gathers the global state and sends it to a remote GPU over dedicated infrastructure" Every ML approach I've heard of including the two you referenced support running on CPU, and the use of a GPU is not an integral part of the bridge pattern. That, or existing frameworks don't at all rely on the bridge pattern as defined here. Note there is some additional confusion because the bridge pattern is defined in software engineering in a way that does not involve GPUs (https://en.wikipedia.org/wiki/Bridge_pattern).

If the issue is actually about a global data gather and not the device architecture, that should be made more clear. Does WRF-ML require a global data gather? It sounds like it interfaces with the model decomposition, but this is unclear to me.

**Removed the term bridge-pattern.**

WRF-ML require a MPI call to reach the surrogate model, which is something that we tried to avoid as it removes any chance for implementations to stay "in-core". We see a scenario where many surrogate models will be called by a GCM with some being kept in-core and others going outside the local rank (for performance reasons or for surrogate models that require spatial information).

L187-188: Does this mean the GPU will contain a separate copy of the ML/AI model in its memory for each rank using that GPU?

**Reply:**

At this stage yes, and only for ranks that are bound to the GPU and only for inference. That should be ok for the use case that we present but not for large DNNs. We will test other architectures once the vectorization feature is ready. E.g. using TorchServe and NVIDIAs multi-processing service.

L253: Since the focus of this paper isn't the ML model itself this isn't a critical issue for the manuscript, but generally speaking it is usually an issue to randomly divide atmospheric model data into training and validation sets. Random sampling results in a high degree of correlation between samples in the training and validation sets, which negates the usefulness of an independent validation dataset. A better split would be to take the last 10% of samples in time as validation. If possible, I would suggest re-training with this split, otherwise including some discussion of the correlation between validation and training samples.

**Reply:**

L253 does not talk about that. The test data was taken from a separate time sample. We have slightly reworded this paragraph to clarify the distinction between training, validation, and evaluation data.

L261-265: It seems that normalization is built into your ML model. Does the framework require the pytorch model handle normalizing inputs/outputs? This isn't bad, but would be important to spell out to potential users of the framework.

**Reply:**

This confusion is seemingly an issue that stems from the reviewer's comment re. L168 (packing and normalizing) which was removed from the text.

Fig 8-9: (Optional) It could be helpful to include contour maps of the scatterplot densities or something similar. The R2 values are surprisingly high given how the figures look, but only because there's no sense of density.

Technical corrections: None.

**Reply:**

We agree that the figures don't show density well, but they are intended to show systematic errors in certain situations that seem more important than the R2 and that may be useful in understanding and improving surrogates in future efforts, and to exhibit the value of the parallel physics operation offered by the plugin. Figs 8 and 9 (7 and 8 in the new manuscript) exhibit some of the less optimal aspects of the surrogate model. For example, the lack of skill in the mid-troposphere and prediction of positive moisture tendencies (Fig. 8). The slope in all the panels is also consistently smaller than 1. and the lack of skill seems related to highly positive tendencies. Some stability issues were resolved by adding constraints at runtime and during the learning stage, but these are clearly not enough.

Reproducibility: To ensure reproducibility, the library https://download.pytorch.org/libtorch/cpu/libtorch-cxx11-abi-shared-with-deps-1.9.0%2Bcpu.zip should be included in the zenodo release unless licensing restrictions prevent this.

**Reply:**

This dependency will be automatically downloaded during the installation. We prefer not to store binaries and encourage the users to use updated versions which will include security and bug fixes.

Some required information is also missing to be able to reproduce building the CESM/CAM reference implementation. In particular, a static link (preferably with a DOI) to where the source code can be downloaded, and a list of libraries and versions used on the NCI infrastructure.

**Reply:**

The reference implementation in CESM/CAM is adjoined in the repository together with the plugin. Note the README.md in the repository (https://github.com/dudek313/torchclim?tab=readme-ov-file#building-the-cesmcam-reference-implementation-with-torchclim).

These details are somewhat important if this is intended to be an open source community project.

Citation: https://doi.org/10.5194/egusphere-2023-1954-RC3

---

## Author Response (AR2)

The following changes were added following the comments made by reviewer 1:

1. Removed the word "stable" from the abstract.
2. Expanded on the issue in the conclusion.